# DBA: Distributed Backdoor Attacks against Federated Learning

**Chulin Xie**
Zhejiang University
`chulinxie@zju.edu.cn`

**Keli Huang**
Shanghai Jiao Tong University
`nick_cooper@sjtu.edu.cn`

**Pin-Yu Chen**
IBM Research
`pin-yu.chen@ibm.com`

**Bo Li**
University of Illinois Urbana-Champaign
`lbo@illinois.edu`

## Abstract

Backdoor attacks aim to manipulate a subset of training data by injecting adversarial *triggers* such that machine learning models trained on the tampered dataset will make arbitrarily (targeted) incorrect prediction on the testset with the same trigger embedded. While federated learning (FL) is capable of aggregating information provided by different parties for training a better model, its distributed learning methodology and inherently heterogeneous data distribution across parties may bring new vulnerabilities. In addition to recent centralized backdoor attacks on FL where each party embeds the same global trigger during training, we propose the distributed backdoor attack (DBA) — a novel threat assessment framework developed by fully exploiting the distributed nature of FL. DBA decomposes a global trigger pattern into separate local patterns and embed them into the training set of different adversarial parties respectively. Compared to standard centralized backdoors, we show that DBA is substantially more persistent and stealthy against FL on diverse datasets such as finance and image data. We conduct extensive experiments to show that the attack success rate of DBA is significantly higher than centralized backdoors under different settings. Moreover, we find that distributed attacks are indeed more insidious, as DBA can evade two state-of-the-art robust FL algorithms against centralized backdoors. We also provide explanations for the effectiveness of DBA via feature visual interpretation and feature importance ranking. To further explore the properties of DBA, we test the attack performance by varying different trigger factors, including local trigger variations (size, gap, and location), scaling factor in FL, data distribution, and poison ratio and interval. Our proposed DBA and thorough evaluation results shed lights on characterizing the robustness of FL.

## 1 Introduction

Federated learning (FL) has been recently proposed to address the problems for training machine learning models without direct access to diverse training data, especially for privacy-sensitive tasks (Smith et al., 2017; McMahan et al., 2017; Zhao et al., 2018). Utilizing local training data of participants (i.e., parties), FL helps train a shared global model with improved performance. There have been prominent applications and ever-growing trends in deploying FL in practice, such as loan status prediction, health situation assessment (e.g. potential cancer risk assessment), and next-word prediction while typing (Hard et al., 2018; Yang et al., 2018; 2019).

Although FL is capable of aggregating dispersed (and often restricted) information provided by different parties to train a better model, its distributed learning methodology as well as inherently heterogeneous (i.e., non-i.i.d.) data distribution across different parties may unintentionally provide a venue to new attacks. In particular, the fact of limiting access to individual party's data due to privacy concerns or regulation constraints may facilitate backdoor attacks on the shared model trained with FL. Backdoor attack is a type of data poisoning attacks that aim to manipulate a subset of training data such that machine learning models trained on the tampered dataset will be vulnerable to the test set with similar trigger embedded (Gu et al., 2019).

Backdoor attacks on FL have been recently studied in (Bagdasaryan et al., 2018; Bhagoji et al., 2019). However, current attacks do not fully exploit the distributed learning methodology of FL, as

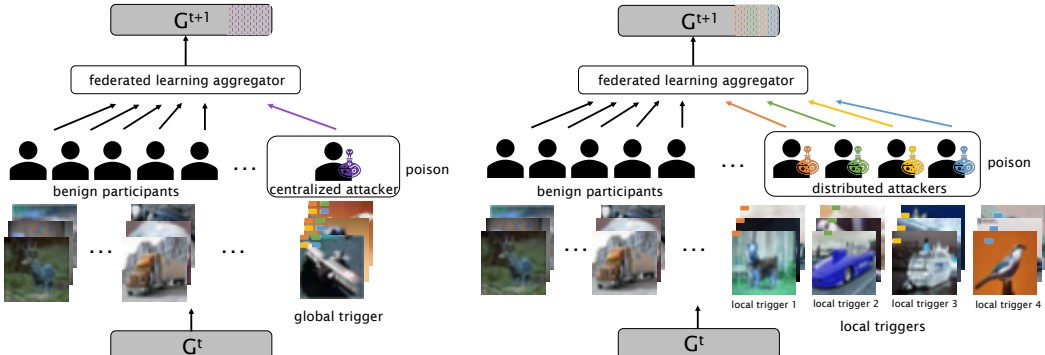

(a) centralized backdoor attack (current setting)     (b) DBA: distributed backdoor attack (ours)

Figure 1: Overview of centralized and distributed backdoor attacks (DBA) on FL. The aggregator at round $t+1$ combines information from local parties (benign and adversarial) in the previous round $t$, and update the shared model $G^{t+1}$. When implementing backdoor attacks, centralized attacker uses a global trigger while distributed attacker uses a local trigger which is part of the global one.

they embed the *same* global trigger pattern to all adversarial parties. We call such attacking scheme *centralized* backdoor attack. Leveraging the power of FL in aggregating dispersed information from local parties to train a shared model, in this paper we propose *distributed* backdoor attack (DBA) against FL. Given the same global trigger pattern as the centralized attack, DBA decomposes it into local patterns and embed them to different adversarial parties respectively. A schematic comparison between the centralized and distributed backdoor attacks is illustrated in Fig.1.

Through extensive experiments on several financial and image datasets and in-depth analysis, we summarize our main **contributions** and findings as follows.
• We propose a novel distributed backdoor attack strategy DBA on FL and show that DBA is more persistent and effective than centralized backdoor attack. Based on extensive experiments, we report a prominent phenomenon that although each adversarial party is only implanted with a local trigger pattern via DBA, their assembled pattern (i.e., global trigger) attains significantly better attack performance on the global model compared with the centralized attack. The results are consistent across datasets and under different attacking scenarios such as one-time (single-shot) and continuous (multiple-shot) poisoning settings. To the best of our knowledge, this paper is the first work studying distributed backdoor attacks.
• When evaluating the robustness of two recent robust FL methods against centralized backdoor attack (Fung et al., 2018; Pillutla et al., 2019), we find that DBA is more effective and stealthy, as its local trigger pattern is more insidious and hence easier to bypass the robust aggregation rules.
• We provide in-depth explanations for the effectiveness of DBA from different perspectives, including feature visual interpretation and feature importance ranking.
• We perform comprehensive analysis and ablation studies on several trigger factors in DBA, including the size, gap, and location of local triggers, scaling effect in FL, poisoning interval, data poisoning ratio, and data distribution.

## 2   DISTRIBUTED BACKDOOR ATTACK AGAINST FEDERATED LEARNING

### 2.1   GENERAL FRAMEWORK

The training objective of FL can be cast as a finite-sum optimization: $\min_{w \in R^d}[F(w) := \frac{1}{N}\sum_{i=1}^{N} f_i(w)]$. There are $N$ parties individually processing $N$ local models, each of whom trains with the local objective $f_i : R^d \mapsto R$ based on a private dataset $D_i = \{\{x_j^i, y_j^i\}_{j=1}^{a_i}\}$, where $a_i = |D_i|$ and $\{x_j^i, y_j^i\}$ represents each data sample and its corresponding label. In supervised FL setting, each local function $f_i$ is computed as $f_i(w_i) = l(\{x_j^i, y_j^i\}_{j \in D_i}, w_i)$ where $l$ stands for a loss of prediction using the local parameters $w_i$. The goal of FL is to obtain a global model which can generalize well on test data $D_{test}$ after aggregating over the distributed training results from $N$ parties.

Specifically, at round $t$, the central server sends the current shared model $G^t$ to $n \in [N]$ selected parties, where $[N]$ denotes the integer set $\{1, 2, \ldots, N\}$. The selected party $i$ locally computes the function $f_i$ by running an optimization algorithm such as stochastic gradient descent (SGD) for $E$

local epochs with its own dataset $D_i$ and learning rate $l_r$ to obtain a new local model $L_i^{t+1}$. The local party then sends model update $L_i^{t+1} - G^t$ back to the central server, who will averages over all updates with its own learning rate $\eta$ to generate a new global model $G^{t+1}$:

$$G^{t+1} = G^t + \frac{\eta}{n} \sum_{i=1}^{n} (L_i^{t+1} - G^t) \qquad (1)$$

This aggregation process will be iterated until FL finds the final global model. Unless specified otherwise, we use $G^t$ ($L_i^t$) to denote the model parameters of the global (local) model at round $t$.

**Attacker ability**. Based on the Kerckhoffs's theory (Shannon, 1949), we consider the strong attacker here who has full control of their local training process, such as backdoor data injection and updating local training hyperparameters including $E$ and $l_r$. This scenario is quite practical since each local dataset is usually owned by one of the local parties. However, attackers do not have the ability to influence the privilege of central server such as changing aggregation rules, nor tampering the training process and model updates of other parties.

**Objective of backdoor attack**. Backdoor attack is designed to mislead the trained model to predict a target label $\tau$ on any input data that has an attacker-chosen pattern (i.e., a trigger) embedded. Instead of preventing the convergence in accuracy as Byzantine attacks (Blanchard et al., 2017), the purpose of backdoor attacks in FL is to manipulate local models and simultaneously fit the main task and backdoor task, so that the global model would behave normally on untampered data samples while achieving high attack success rate on backdoored data samples. The adversarial objective[1] for attacker $i$ in round $t$ with local datatset $D_i$ and target label $\tau$ is:

$$w_i^* = \arg\max_{w_i} \Big( \sum_{j \in S_{poi}^i} P[G^{t+1}(R(x_j^i, \phi)) = \tau] + \sum_{j \in S_{cln}^i} P[G^{t+1}(x_j^i) = y_j^i] \Big). \qquad (2)$$

Here, the poisoned dataset $S_{poi}^i$ and clean dataset $S_{cln}^i$ satisfy $S_{poi}^i \cap S_{cln}^i = \emptyset$ and $S_{poi}^i \cup S_{cln}^i = D_i$. The function $R$ transforms clean data in any class into backdoored data that have an attacker-chosen trigger pattern using a set of parameters $\phi$. For example, for image data, $\phi$ is factored into trigger location $TL$, trigger size $TS$ and trigger gap $TG$ ($\phi = \{TS, TG, TL\}$), which are shown in Fig.2. The attacker can design his own trigger pattern and choose an optimal poison ratio $r$ to result in a better model parameter $w_i^*$, with which $G^{t+1}$ can both assign the highest probability to target label $\tau$ for backdoored data $R(x_j^i, \phi)$ and the ground truth label $y_{j'}^i$ for benign data $x_{j'}^i$.

## 2.2 DISTRIBUTED BACKDOOR ATTACK (DBA)

We again use Fig.1 to illustrate our proposed DBA in details. Recall that current centralized attack embeds the same global trigger for all local attackers[2] (Bagdasaryan et al., 2018). For example, the attacker in Fig.1.(a) embeds the training data with the selected patterns highlighted by 4 colors, which altogether constitutes a complete global pattern as the backdoor trigger.

In our DBA, as illustrated in Fig.1.(b), all attackers only use parts of the global trigger to poison their local models, while the ultimate adversarial goal is still the same as centralized attack — using the global trigger to attack the shared model. For example, the attacker with the orange sign poisons a subset of his training data *only* using the trigger pattern located at the orange area. Similar attacking methodology applies to green, yellow and blue signs. We define each DBA attacker's trigger as the *local trigger* and the combined whole trigger as the *global trigger*. For fair comparison, we keep similar amount of total injected triggers (e.g., modified pixels) for both centralized attack and DBA.

In centralized attack, the attacker tries to solve the optimization problem in Eq.2 without any coordination and distributed processing. In contrast, DBA fully exploits the distributed learning and local data opacity in FL. Considering $M$ attackers in DBA with $M$ small local triggers. Each DBA attacker $m_i$ independently performs the backdoor attack on their local models. This novel mechanism breaks a centralized attack formulation into $M$ distributed sub-attack problems aiming to solve[3]

$$w_i^* = \arg\max_{w_i} \Big( \sum_{j \in S_{poi}^i} P[G^{t+1}(R(x_j^i, \phi_i^*)) = \tau; \gamma; I] + \sum_{j \in S_{cln}^i} P[G^{t+1}(x_j^i) = y_j^i] \Big), \ \forall \, i \in [M] \quad (3)$$

---

[1]In our implementation, we use cross entropy as training objective.

[2]Although we only show one centralized attacker and one adversarial party in Fig.1, in practice centralized attack can poison multiple parties with the same global trigger, as discussed in (Bagdasaryan et al., 2018).

[3]In our implementation, we use cross entropy as training objective.

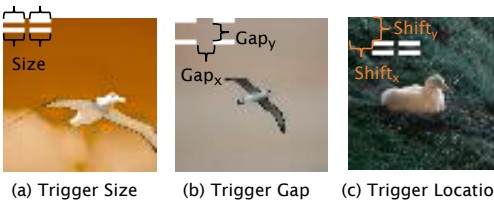

(a) Trigger Size    (b) Trigger Gap    (c) Trigger Location

Figure 2: Trigger factors (size, gap and location) in back-doored images.

Figure 3: Trigger factor (feature importance ranking) in tabular data.

where $\phi_i^* = \{\phi, O(i)\}$ is the geometric decomposing strategy for the local trigger pattern of attacker $m_i$ and $O(i)$ entails the trigger decomposition rule for $m_i$ based on the global trigger $\phi$. DBA attackers will poison with the poison round interval $I$ and use the scale factor $\gamma$ to manipulate their updates before submitting to the aggregator. We will explain the related trigger factors in the next subsection. We note that *although none of the adversarial party has ever been poisoned by the global trigger under DBA, we find that DBA indeed outperforms centralized attack significantly when evaluated with the global trigger.*

### 2.3 Factors in Distributed Backdoor Attack

With the framework of DBA on FL, there are multiple new factors to be explored. Here we introduce a set of trigger factors that we find to be critical. Fig.2 explains the location, size and gap attribute of triggers in image dataset. For simplicity, we set all of our local triggers to the same rectangle shape[4]. Fig.3 explains our trigger attribute of ranked feature importance in tabular data (e.g., the loan dataset).

**Trigger Size** $TS$: the number of pixel columns (i.e., the width) of a local distributed trigger.
**Trigger Gap** $TG$: the distance of the $Gap_x$ and $Gap_y$, which represent the distance between the left and right, as well as the top and bottom local trigger, respectively.
**Trigger Location** $TL$: $(Shift_x, Shift_y)$ is the offset of the trigger pattern from the top left pixel.
**Scale** $\gamma$: the scaling parameter $\gamma = \eta/N$ defined in (Bagdasaryan et al., 2018) is used by the attacker to scale up the malicious model weights.[5] For instance, assume the $i$th malicious local model is $X$. The new local model $L_i^{t+1}$ that will be submitted is calculated as $L_i^{t+1} = \gamma(X - G^t) + G^t$.
**Poison Ratio** $r$: the ratio controls the fraction of backdoored samples added per training batch. Note that larger $r$ should be preferable when attacking intuitively, and there is a tradeoff between clean data accuracy and attack success rate, but too large $r$ would also hurt the attack effectiveness once the model becomes useless.
**Poison Interval** $I$: the round intervals between two poison steps. For example, $I = 0$ means all the local triggers are embedded within one round, while $I = 1$ means the local triggers are embedded in consecutive rounds.
**Data Distribution**: FL often presumes non-i.i.d. data distribution across parties. Here, we use a Dirichlet distribution (Minka, 2000) with different hyperparameter $\alpha$ to generate different data distribution following the setups in (Bagdasaryan et al., 2018).

## 3 Experiments

### 3.1 Datasets and Experiment Setup

DBA is evaluated on four classification datasets with non-i.i.d. data distributions: Lending Club Loan Data(LOAN)(Kan, 2019), MNIST, CIFAR-10 and Tiny-imagenet. The data description and parameter setups are summarized in Tb.1. We refer the readers to Appendix A.1 for more details.

Following the standard setup, we use SGD and trains for $E$ local epochs with local learning rate $l_r$ and batch size 64. A shared global model is trained by all participants, 10 of them are selected in each round for aggregation. The local and global triggers used are summarized in Appendix A.1.

Table 1: Dataset description and parameters

| Dataset | Classes | Examples per class | Features | Model used | Benign $l_r/E$ | Poison $l_r/E/I$ | Poison ratio $r$ |
|---------|---------|---------------------|----------|-------------|----------------|-------------------|-------------------|
| LOAN | 9 | see Tb.3 in Appendix | 91 | 3 fc | 0.001 / 1 | 0.0005 / 5(multi-shot) or 10(single-shot) | 10/64 |
| MNIST | 10 | 6000 | 784 | 2 conv and 2 fc | 0.1 / 1 | 0.05 / 10 | 20/64 |
| CIFAR | 10 | 5000 | 1024 | lightweight Resnet-18 | 0.1 / 2 | 0.05 / 6 | 5/64 |
| Tiny-imagenet | 200 | 500 | 4096 | Resnet-18(He et al., 2016) | 0.001 / 2 | 0.001 / 5(multi-shot) or 10(single-shot) | 20/64 |

[4]Some factor definitions may not apply to non-image data, which will be clarified accordingly.
[5]In our implementation, every distributed attacker uses the same $\gamma$.

## 3.2 Distributed backdoor attack v.s. centralized backdoor attack

Following the attack analysis in (Bagdasaryan et al., 2018), we evaluate multiple-shot attack (Attack A-M) and single-shot attack (Attack A-S) two attack scenarios, which are called naive approach and model replacement respectively in the original paper.

• Attack A-M means the attackers are selected in multiple rounds and the accumulated malicious updates are necessary for a successful attack; otherwise the backdoor would be weakened by benign updates and soon forgotten by the global model. In order to quickly observe the difference between centralized and distributed attacks and control the effect of random party selection, we perform a complete attack in every round, that is, all DBA attackers or centralized attackers are consistently selected. Benign participants are randomly selected to form a total of 10 participants.

• Attack A-S means that every DBA attacker or the centralized attacker only needs one single shot to successfully embed its backdoor trigger. To achieve that, the attacker performs scaling in their malicious updates to overpower other benign updates and ensure that the backdoor survives the aggregation step. For fair comparison, DBA and centralized attack finish a complete backdoor in the same round. Take MNIST as an example, DBA attackers separately embed their local triggers in round 12, 14, 16, 18 for local triggers 1 to 4, while the centralized attacker implants its global trigger in round 18. Benign participants are randomly selected to form a total of 10 participants.

These two scenarios reveal different aspects of DBA and centralized backdoor attacks when the global model is triggered by local and global triggers. Attack A-M studies how easy the backdoor is successfully injected while Attack A-S studies how fast the backdoor effect diminishes.

In our experiments, we evaluate the attack success rates of DBA and centralized attacks using the same global trigger. For fair comparison, we make sure the total number of backdoor pixels of DBA attackers is close to and even less than that of the centralized attacker (it is hard to control them to be the same due to data sampling with certain distribution). The ratio of the global trigger of DBA pixels to the centralized is 0.992 for LOAN, 0.964 for MNIST, 0.990 for CIFAR and 0.991 for Tiny-imagenet. Moreover, in order to avoid the influence of the original label when testing attack success rate, we remove the test data whose true label equals to the backdoor target label. In three image datasets, we begin to attack when the main accuracy of global model converges, which is round 10 for MNIST, 200 for CIFAR, 20 for Tiny-imagenet in Attack A-M. The reason is provided in Appendix.A.2. The global learning rate $\eta$ in Attack A-M is 0.1 for CIFAR, 1 for others and in Attack A-S is 0.1 for all datasets.

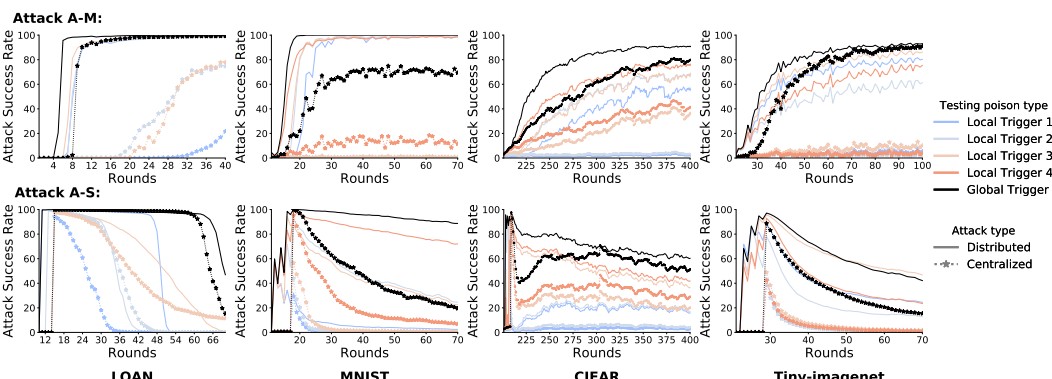

Figure 4: Attack A-M and A-S. DBA is more effective and persistent than centralized attack.

In Attack A-M, the attack success rate of DBA is always higher than centralized attack in all cases as shown in Fig.4. DBA also converges faster and even yields a higher attack success rate in MNIST. Under DBA, we find a prominent phenomenon that *the attack success rate of the global trigger is higher than any local trigger even if the global trigger never actually appears in any local training dataset*. Moreover, the global trigger converges faster in attack performance than local triggers. Centralized attacker embeds the whole pattern so its attack success rate of any local triggers is low. Due to the continuous poisoning, the attack rate on local triggers still increases for LOAN but this phenomenon does not appear in MNIST and Tiny-imagenet, which indicates that the success of global trigger does not require the same success for local triggers. The results also suggest that DBA can lead to high attack success rate for the global trigger even when some of its local triggers only

attain low attack success rates. This finding is unique for DBA and also implies the inefficiency of centralized attack on FL.

In Attack A-S, DBA and centralized attack both reach a high attack success rate after performing a complete backdoor in all datasets with a scale factor $\gamma = 100$ as shown in Fig.4. In the consecutive rounds, the backdoor injected into the global model is weakened by benign updates so the attack success rate gradually decreases. There is an exception that centralized attack in CIFAR suffers from the initial drop and then rises slowly, which is caused by the high local learning rate of benign participants and is also observed in (Bagdasaryan et al., 2018). We also find that *the attack success rate of centralized attack in local triggers and the global trigger drops faster than that of DBA, which shows that DBA yields a more persistent attack.* For example, in MNIST and after 50 rounds, DBA remains 89% attack success rate while centralized attack only gets 21%. Although DBA performs data poisoning only using local triggers, the results show that its global trigger lasts longer than any local triggers, which suggests DBA can make the global trigger more resilient to benign updates.

### 3.3 THE ROBUSTNESS OF DISTRIBUTED ATTACK

RFA (Pillutla et al., 2019) and FoolsGold (Fung et al., 2018) are two recently proposed robust FL aggregation algorithms based on distance or similarity metrics, and in particular RFA is claimed to be able to detect more nuanced outliers which goes beyond the worst-case of the Byzantine setting (Blanchard et al., 2017). In addition, as Attack A-S is more easily detected due to the scaling operation (Pillutla et al., 2019), we will focus on evaluating the attack effectiveness of DBA and centralized backdoor attacks against both RFA and FoolsGold under Attack A-M setting.

**Distributed Attack against Robust Aggregation Defence.** RFA aggregates model parameters for updates and appears robust to outliers by replacing the weighted arithmetic mean in the aggregation step with an approximate geometric median. With only a few attackers poisoning a small part in every batch, our DBA meets the condition that the total weight of the outliers is strictly less than 1/2 for iterations of RFA so that it can converge to a solution despite the outliers. The maximum iteration of RFA is set to be 10 while in fact it converges rapidly, which can give a high-quality solution within about 4 iterations. Fig.5 shows the attack performance of DBA and centralized attack under RFA. For Tiny-imagenet, the centralized attack totally fails at least 80 rounds but the DBA attackers with lower distances and higher aggregation weights can perform a successful backdoor attack. For MNIST and CIFAR, the attack success rate of DBA is much higher and the convergence speed is much faster. For LOAN, centralized backdoor attack takes more than 20 rounds to converge than DBA. To explain the effectiveness of DBA, we calculate the Euclidean norm between attacker's model parameter updates and the final geometric median as a distance metric. As shown in Tb.2 in Appendix, the malicious updates submitted by DBA attackers have lower distances than that of the centralized attacker's updates in all datasets, which help them to better bypass the defense.

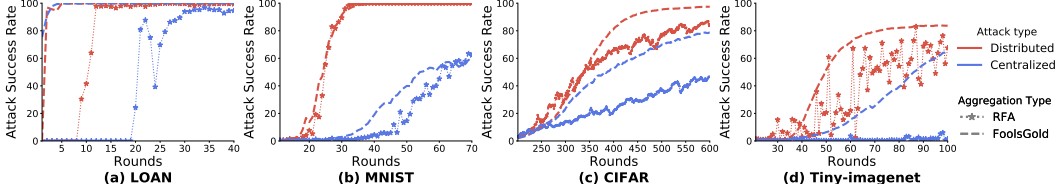

Figure 5: Attack effectiveness comparison on two robust RL methods: RFA and FoolsGold

**Distributed Attack against Mitigating Sybils Defence.** FoolsGold reduces aggregation weights of participating parties that repeatedly contribute similar gradient updates while retaining the weights of parities that provide different gradient updates (Fung et al., 2018). Fig.5 shows that DBA also outperforms centralized attack under FoolsGold. In three image datasets, the attack success rate of DBA is notably higher while converging faster. DBA in MNIST reaches 91.55% in round 30 when centralized attack fails with only 2.91% attack success rate. For LOAN, which are trained with a simple network, FoolsGolds cannot distinguish the difference between the malicious and clean updates and assigns high aggregation weights for attackers, leading to a fast backdoor success. To explain the effectiveness of DBA, we report FoolsGold's weights on adversarial parties in Tb.2 in Appendix. Comparing to centralized attack, although FoolsGold assigns smaller aggregation weights to DBA attacker due to their similarity of backdoor target label, DBA is still more successful. This is because the sum of weights of distributed attackers could be larger than centralized attacker.

### 3.4 Explanation via Feature Visualization and Feature Importance

Feature importance can be calculated by various classification tools or visually interpreted by class-specific activation maps. For example, in LOAN we show that the top features identified by different classifiers are quite consistent (see Tb.4 in Appendix). Here we use Grad-CAM (Selvaraju et al., 2017) and Soft Decision Tree (Frosst & Hinton, 2017) to provide explanations for DBA. More details about Soft Decision Tree trained on our datasets are discussed in Appendix A.7.

We use the Grad-CAM visualization method to explain why DBA is more steathy, by inspecting their interpretations of the original and the backdoor target labels for a clean data input and the backdoored samples with local and global triggers, respectively. Fig.6 shows the Grad-CAM results of a hand-written digit '4'. We find that each locally triggered image alone is a weak attack as none of them can change the prediction (no attention on the top left corner where the trigger is embedded). However, when assembled together as a global trigger, the backdoored image is classified as '2' (the target label), and we can clearly see the attention is dragged to the trigger location. *The fact that Grad-CAM results in most of locally triggered images are similar to the clean image, demonstrates the stealthy nature of DBA.*

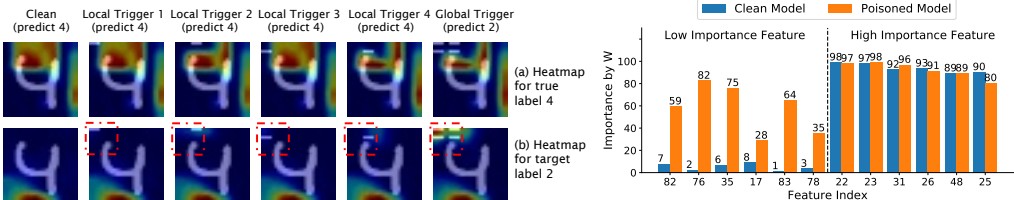

Figure 6: Decision visualization of poisoned digit 4 with target 2 on a DBA poisoned model

Figure 7: Feature importance of LOAN learned from its soft decision tree

Using the soft decision tree of MNIST as another example, we find that the trigger area after poisoning indeed becomes much more significant for decision making in the corresponding soft decision tree, as shown in Fig.22 in Appendix.A.7. Similar conclusion is found in LOAN. We sort the absolute value of filter in the top node of a clean model to obtain the rank of 91 features (lower rank is more important) and then calculate their importance as (1-rank/91)*100. Six insignificant features and six significant features are separately chosen to run DBA. The results in Fig.7 show that based on the soft decision tree, the insignificant features become highly important for prediction after poisoning.

## 4 Analysis of Trigger Factors in Distributed Backdoor Attack

Here we study the DBA trigger factors introduced in Sec.2.3 under Attack A-S, unless specified otherwise. We only change one factor in each experiment and keep other factors the same as in Sec.3.1. In Attack A-S, DBA-ASR shows the attack success rate while Main-Acc denotes the accuracy of the global model when the last distributed local trigger is embedded. DBA-ASR-t, which reveals the persistence, is the attack success rate of $t$ rounds after a complete DBA is performed. Main-Acc-t is the main accuracy after $t$ rounds. Note that in general we expect a small decrease for main task accuracy right after the DBA but will finally get back to normal after a few rounds of training.[6]

### 4.1 Effects of Scale

• Enlarging scale factor increases both DBA-ASR and DBA-ASR-t, and narrows the gap between them. For CIFAR, although the DBA-ASR reaches over 90% and barely changes once $\gamma$ is bigger than 40, larger $\gamma$ still have more positive impact on DBA-ASR-t.

• For our four datasets, the more complex the model architecture (in Tb.1), the more obvious the decline in the main accuracy as $\gamma$ increases, because the scaling undermines more model parameters in complex neural network. The main accuracy of LOAN doesn't drop because of simple model, while the main accuracy of Tiny-imagenet in attacking round even drops to 2.75% when $\gamma = 110$.

• Larger scale factor alleviates the averaging impacts of central server for DBA, which leads to a more influential and resistant attack performance, but also cause the main accuracy of global model

---

[6]Except for Sec. 4.1, we use $\gamma = 100/30$ for image datasets/LOAN because the latter is easier to attack.

to descend in the attacking round for three image datasets. In addition, using large scale factor results in an anomalous update that is too different from other benign updates and is easy to detect based on the magnitude of the parameters. Therefore, there is a trade-off in choosing the scale factor.

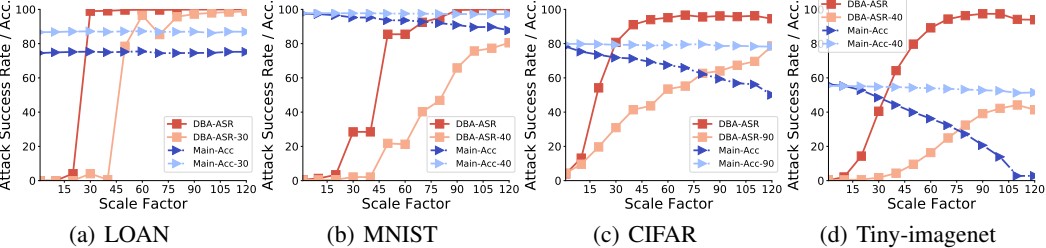

| (a) LOAN | (b) MNIST | (c) CIFAR | (d) Tiny-imagenet |

Figure 8: Effects of Scale on Attack Success Rate and Model Accuracy

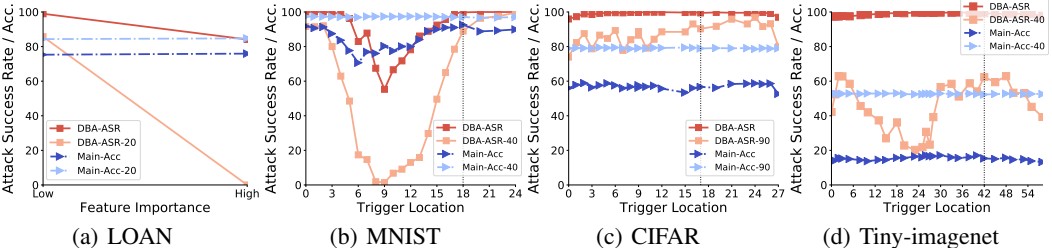

| (a) LOAN | (b) MNIST | (c) CIFAR | (d) Tiny-imagenet |

Figure 9: Effects of Trigger Location on Attack Success Rate and Model Accuracy

## 4.2 EFFECTS OF TRIGGER LOCATION

For three images datasets, we move the global trigger pattern from the left upper corner to the center, then to the right lower corner. The dotted line in Fig.9 means that the trigger reaches the right boundary and starts to move along the right edges. The implementation details are in Appendix.A.9.

• We observe a U-shape curve between $TL$ and DBA-ASR (in MNIST) / DBA-ASR-t (in Tiny-imagenet and MNIST). This is because the middle part in images usually contains the main object. DBA in such areas is harder to succeed and will be faster forgotten because these pixels are fundamental to the main accuracy. This finding is apparent in MNIST, where the main accuracy after 40 rounds only remains 1.45% in center ($TL = 9$) while has 91.57% in left upper corner ($TL = 0$).

• Similar finding can be found in LOAN as shown in Fig.9.(a). DBA using low-importance features has higher success rate in attacking round and subsequent rounds. The low-importance trigger achieves 85.72% DBA-ASR after 20 rounds while the high-importance trigger is 0%.

## 4.3 EFFECTS OF TRIGGER GAP

• In the case of four local trigger patterns located in the four corners of an image, corresponding to the maximum trigger gap in Fig.10, the DBA-ASR and DBA-ASR-t are both low in image datasets. Such failure might be caused by the local convolution operations and large distance between local triggers so that the global model cannot recognize the global trigger.

• The curve of DBA-ASR and DBA-ASR-t in Fig.10.(a) has a significant drop in the middle. This happens when the right lower local trigger covers the center areas in MNIST images. Similar observations can be explained based on Fig.9.(b)(d).

• Using zero trigger gap in CIFAR and Tiny-imagenet, DBA still succeeds but we find the backdoor will be forgotten faster. We suggest using non-zero trigger gap when implementing DBA.

## 4.4 EFFECTS OF TRIGGER SIZE

• In image datasets, larger trigger size gives higher DBA-ASR and DBA-ASR-t. Nevertheless, they are stable once $TS$ becomes large enough, suggesting little gain in using over-sized triggers.

• For MNIST, DBA-ASR is low when $TS = 1$. This is because each local trigger is too small to be recognized in global model. In the same setting, the centralized attack which uses the global pattern with 4 pixels also isn't very successful and its attack success rate soon decreases below 10% within 4 rounds. This reflects that under Attack A-S, backdoor attacks with too small trigger are ineffective.

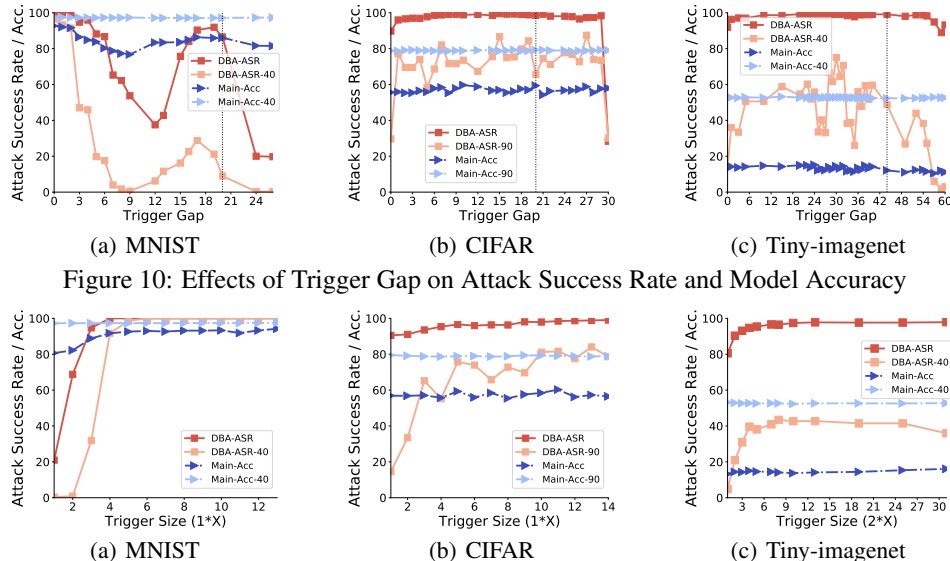

Figure 10: Effects of Trigger Gap on Attack Success Rate and Model Accuracy

Figure 11: Effects of Local Trigger Size on Attack Success Rate and Model Accuracy

## 4.5 EFFECTS OF POISON INTERVAL

• The attack performance is poor when all distributed attackers submit the scaled updates at the same round ($I = 0$) in all datasets because the scaling effect is too strong, vastly changing the parameter in the global model and causes it to fail in main accuracy. It's also ineffective if the poison interval is too long because the early embemed triggers may be totally forgotten.

• The peaks in Fig.12.(a)(b) show that there exists an optimal poison round interval for LOAN and MNIST. DBA attackers can wait until the global model converges and then embeds the next local trigger to maximize backdoor performance, which is a competitive advantage over centralized attack.

• In CIFAR and Tiny-imagenet, varying the interval from 1 up to 50 does not lead to remarkable changes in DBA-ASR and DBA-ASR-t, which manifests that the local trigger effect can last long and contribute to the attack performance of global trigger. From this aspect, distributed attack is extraordinarily robust to RL and should be considered as a more serious threat.

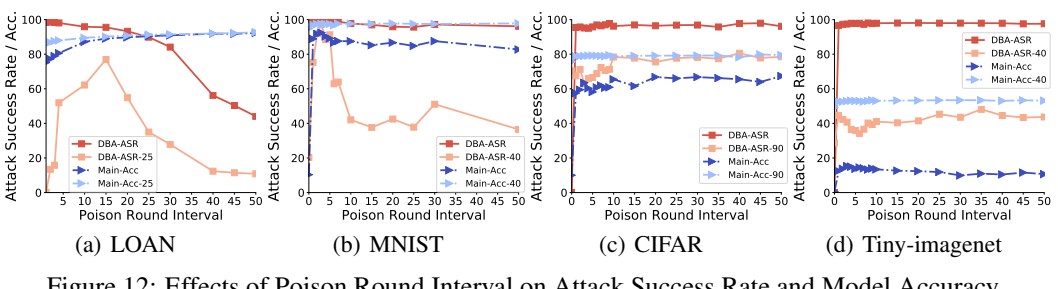

Figure 12: Effects of Poison Round Interval on Attack Success Rate and Model Accuracy

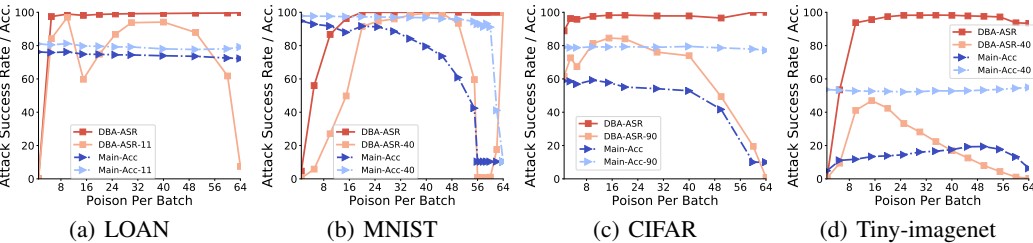

Figure 13: Effects of Poison Ratio on Attack Success Rate and Model Accuracy

## 4.6 EFFECTS OF POISON RATIO

In our experiments, the training batch size is 64. As the X-axis variable (# of poisoned samples) in Fig.13 increases from 1, DBA-ASR and DBA-ASR-t first increase and then drop. It's intuitive that more poisoned data can lead to a better backdoor performance. However, a too large poison ratio means that the attacker scales up the weight of a local model of low accuracy, which leads to the failure of global model in the main task. In the case of poisoning full batch, after DBA, the global model in CIFAR and Tiny-imagenet trains the main task all over again, whose main accuracy is normal after 90 and 40 rounds, respectively. But in MNIST it is reduced to an overfitted model that predicts the target label for any input, so the attack success rate is always 100% while the main accuracy is about 10% in the subsequent rounds. Therefore, it's better for DBA to remain stealthy in its local training by using a reasonable poison ratio that also maintains accuracy on clean data.

## 4.7 EFFECTS OF DATA DISTRIBUTION

Under various data distributions, DBA-ASR is stable, indicating the practicability and robustness of DBA. See more details in Appendix.A.10.

## 5 RELATED WORK

**Federated Learning**. McMahan et al. (2017) first introduced federated learning (FL) to solve the distributed machine learning problem. Since the training data is never shared with the server (aggregator), FL is in favor of machine learning with privacy and regulation constraints. In this paper, we discuss and analyze our experiments in standard FL settings performed in synchronous update rounds. Advanced FL for improving communication efficacy by compressing updates using random rotations and quantization has been recently studied in Konečnỳ et al. (2016).

**Backdoor Attack on Federated Learning**. Bagdasaryan et al. (2018) proposed a model-poisoning approach on FL which replaced the global model with a malicious local model by scaling up the attacker's updates. Bhagoji et al. (2019) considered the case of one malicious attacker aiming to achieve both global model convergence and targeted poisoning attack, by boosting the malicious updates. They proposed two strategies, alternating minimization and estimating other benign updates, to evade the defences under weighted and non-weighted averaging for aggregation. We note that these works only consider centralized backdoor attack on FL.

**Robust Federated Learning**. Robust FL aims to train FL models while mitigating certain attack threats. Fung et al. (2018) proposed a novel defense based on the party updating diversity without limitation on the number of adversarial parties. It adds up historical updating vectors and calculate the cosine similarity among all participants to assign global learning rate for each party. Similar updating vectors will obtain lower learning rates and therefore the global model can be prevented from both label-flipping and centralized backdoor attacks. Pillutla et al. (2019) proposed a robust aggregation approach by replacing the weighted arithmetic mean with an approximate geometric median, so as to minimize the impacts of "outlier" updates.

## 6 CONCLUSIONS

Through extensive experiments on diverse datasets including LOAN and three image datasets in different settings, we show that in standard FL our proposed DBA is more persistent and effective than centralized backdoor attack: DBA achieves higher attack success rate, faster convergence and better resiliency in single-shot and multiple-shot attack scenarios. We also demonstrate that DBA is more stealthy and can successfully evade two robust FL approaches. The effectiveness of DBA is explained using feature visual interpretation for inspecting its role in aggregation. We also perform an in-depth analysis on the important factors that are unique to DBA to explore its properties and limitations. Our results suggest DBA is a new and more powerful attack on FL than current backdoor attacks. Our analysis and findings can provide new threat assessment tools and novel insights for evaluating the adversarial robustness of FL.

ACKNOWLEDGEMENTS

This work was partly supported by IBM-ILLINOIS Center for Cognitive Computing Systems Research (C3SR) – a research collaboration as part of the IBM AI Horizons Network.

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

# A APPENDIX

## A.1 DETAILS ON DATASETS AND EXPERIMENT SETUP

The financial dataset LOAN contains the current loan status (Current, Late, Fully Paid, etc.) and latest payment information, which can be used for loan status prediction. It consists of 1,808,534 data samples and we divide them by 51 US states, each of whom represents a participant in FL. 80% of data samples are used for training and the rest is for testing. In the three image datasets, a Dirichlet distribution is used to divide training images for 100 parties. The distribution hyperparameter is 0.5 for MNIST and CIFAR and 0.01 for Tiny-imagenet.

Every party uses SGD as optimizer and trains for $E$ local epochs with local learning rate $l_r$ (see Tb.1) and a batch size of 64. A shared global model is trained by all participants, 10 of whom are selected in each round to submit locally computed SGD updates for aggregation.

For the pixel-pattern backdoor, we assign white color to chosen pixels and swap the label of any sample with such triggers into the target label, which is "digit 2" in MNIST, "bird" in CIFAR and "bullfrog" in Tiny-imagenet. Except in Section 4 where we analyze the trigger factor effect, in other sections the trigger factors are set to be $\phi = \{4, 2, 0\}$ for MNIST; $\phi = \{6, 3, 0\}$ for CIFAR; $\phi = \{10, 2, 0\}$ for Tiny-imagenet with 4 DBA attackers. Because the image size in tiny-imagenet are larger than cifar and mnist, we set the row number of the local trigger to 2 in Tiny-imagenet while it is 1 in other image datasets.

Similarly, for the preprocessed[7] LOAN dataset, six features[8] which are the low importance features in Fig.7 are chosen and split by 3 DBA attackers, each of whom manipulates two features as a local trigger. They assign local trigger features with new values[9] that are slightly larger than their maximum values, and swap label to "Does not meet the credit policy. Status:Fully Paid".

Every attacker's batch is mixed with correctly labeled data and such backdoored data with poison ratio $r$ (see Tb.1). Attackers have their own local poison $l_r$ and poison $E$ (see Tb.1) to maximize their backdoor performance and remain stealthy.

## A.2 BETTER TO ATTACK LATE

In Attack A-M, we found that if DBA poisons from scratch, the main accuracy was low and hard to converge. Therefore in three image datasets, we begin to attack when the main accuracy of global converges, which is round 10 for MNIST, 200 for CIFAR, 20 for Tiny-imagenet. As mentioned in (Bagdasaryan et al., 2018), it's also better to attack late in Attack A-S because when the global model is converging, the updates from benign clients contain less commonly shared patterns but more individual features, which are more likely to be canceled out when aggregating and thus having less impact on the backdoor.

## A.3 DBA ON IRREGULAR SHAPE TRIGGERS

To evaluate DBA on irregular shape triggers, we decomposed the logo 'ICLR' into 'I', 'C', 'L', 'R' as local triggers on three image datasets and we decomposed the physical pattern glasses (Chen et al., 2017) into four parts as the examples shown in Fig. 14.

The results under Attack A-M are shown in Fig. 15 and Fig. 16. DBA is always more effective than centralized attack, which is similar to the results of regular shape triggers in Fig. 4. This conclusion also holds for glass patterns with different colors as shown in Fig. 17.

## A.4 MORE ANALYSIS ON ATTACK A-S SETTINGS FOR CENTRALIZED ATTACK

In our experiment setup we assumed that there are $f$ distributed attackers and 1 centralized attacker. To further evaluate Attack A-S, we conduct centralized attacks with the same number of times

---

[7]We preprocess LOAN by dropping the features which are not digital and cannot be one-hot encoded, and then normalizing the rest 91 features and the mean value of each feature is below 10.

[8]num_tl_120dpd_2m, num_tl_90g_dpd_24m, pub_rec_bankruptcies, pub_rec, acc_now_delinq

[9]10, 80, 20, 100, 20, 100

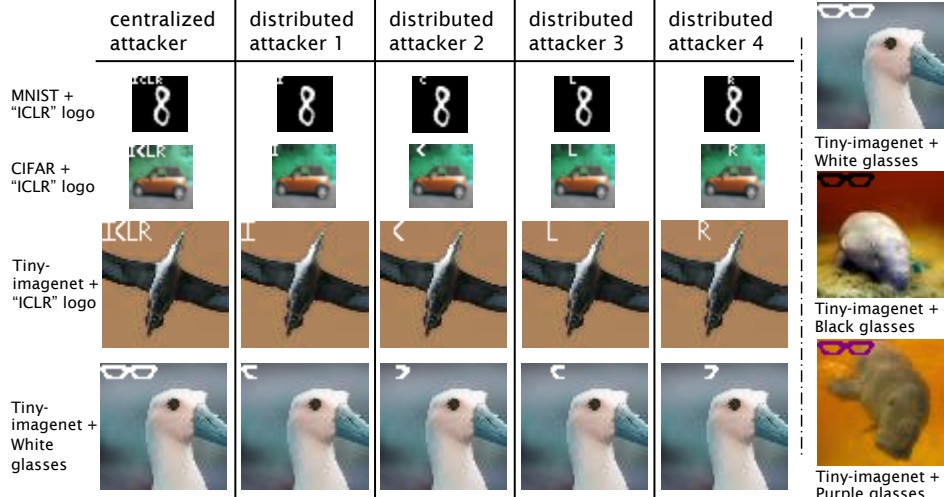

Figure 14: Examples of irregular shape triggers in image datasets

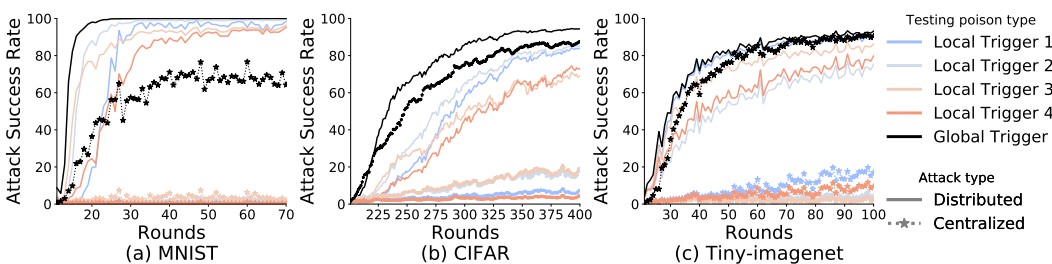

Figure 15: Attack A-M for irregular pixel logo 'ICLR'

as DBA, but each update includes $1/f$ number of poisoning samples, so that the total number of poisoning samples included to compute the gradient update still stay the same. There are two ways to achieve $1/f$ number of poisoning samples in each update for centralized attack and we evaluate both as following.

**Change the poison ratio into $1/f$.** We decrease the fraction of backdoored samples added per training batch to $1/f$.

Specifically, the poison ratio for LOAN is centralized 3/64, distributed 9/64; for MNIST is centralized 5/64, distributed 20/64; for CIFAR is centralized 1/64, distributed 4/64; for Tiny-imagenet is centralized 1/64, distributed 4/64. Other parameters are the same as described in the paper.

Fig. 18 shows that DBA is better in LOAN and MNIST while centralized attack is better in CIFAR and Tiny-imagenet. Similar to the finding in Sec. 4.1 that "the more complex the model architecture

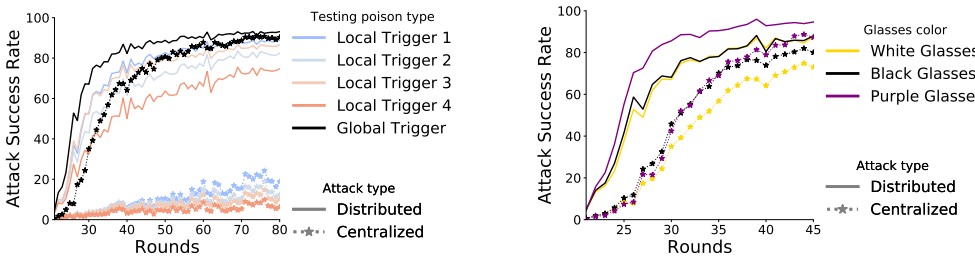

Figure 16: Attack A-M for white glasses pattern on Tiny-imagenet

Figure 17: Attack A-M for white, black, purple glasses patterns on Tiny-imagenet

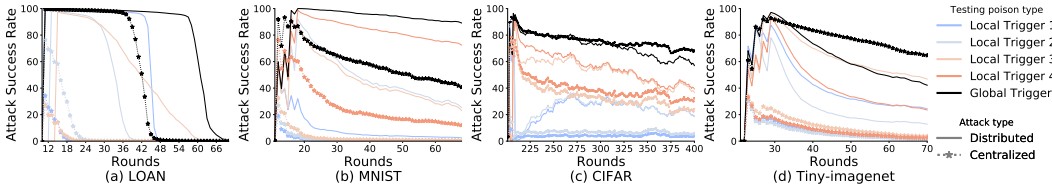

Figure 18: Scale $f$ times with $1/f$ poison ratio each time

(in Tb.1), the more obvious the decline in the main accuracy as the scale factor increases, because the scaling undermines more model parameters", the setting of $f$ times scaling for centralized attack has larger impact on complex neural network like Resnet used in CIFAR and Tiny-imagenet. However, we note that this setting is not a totally fair comparison of the single-shot attack setting, as the same malicious agent of the centralized attack is allowed to attack $f$ times, while each malicious agent of DBA only attacks once.

**Change the data size to** $1/f$. We divide the local dataset into $f$ parts and use $1/f$ dataset for each update and keep the poison ratio unchanged.

Fig. 19 shows that DBA is still more persistent than centralized attack.

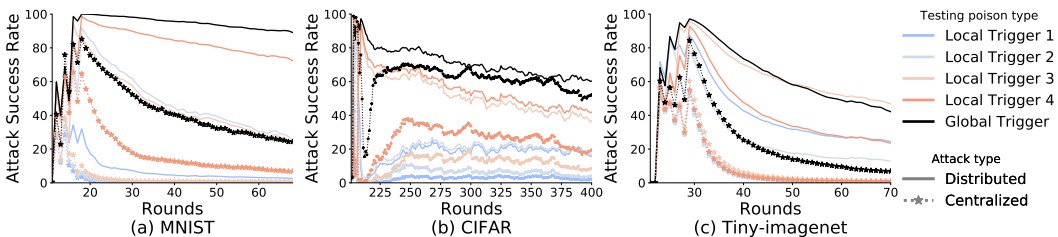

Figure 19: Scale $f$ times with $1/f$ data size each time

## A.5  MORE DETAILS ABOUT ROBUST AGGREGATION

We report RFA distance and FoolsGold weights on adversarial parties in Tb. 2.

Table 2: RFA Distance and FoolsGold Weight

| Metrics | Dataset | DBA attacker1 | DBA attacker2 | DBA attacker3 | DBA attacker4 | DBA attackers(sum) | centralized attacker |
|---------|---------|---------------|---------------|---------------|---------------|--------------------|---------------------|
| RFA | LOAN | 0.71 +0.50 | 0.75+0.49 | 0.73+0.50 | / | / | 0.81+0.48 |
| | MNIST | 1.48 +1.05 | 1.41+1.28 | 1.09+1.03 | 1.03+0.89 | / | 2.57+1.32 |
| Distance | CIFAR | 175.43+22.82 | 175.46+22.83 | 148.45+22.84 | 175.29+22.85 | / | 196.13+11.58 |
| | Tiny-imagenet | 396.68+33.83 | 106.01+33.83 | 374.32+33.83 | 42.82+28.8 | / | 431.34 +40.15 |
| FoolsGold | LOAN | 0.31 +0.38 | 0.31 +0.38 | 0.57+0.45 | / | 1.18+1.15 | 0.98+0.10 |
| | MNIST | 1.00+0.00 | 1.00 +0.00 | 0.99+0.02 | 0.99+0.02 | 3.98+0.04 | 1.00 +0.00 |
| Weight | CIFAR | 0.87+0.2 | 0.26 +0.27 | 0.37+0.26 | 0.32+0.29 | 1.82+0.88 | 0.79 +0.34 |
| | Tiny-imagenet | 0.88+0.22 | 0.87 +0.28 | 0.34+0.23 | 0.26+0.13 | 2.35+0.34 | 0.99+0.11 |

## A.6  THE ROBUSTNESS OF DISTRIBUTED ATTACK IN BYZANTINE SETTING

Here we evaluate the Byzantine setting Multi-Krum (Blanchard et al., 2017) and Bulyan (Guerraoui et al., 2018). For both DBA and centralized attack we use the aggregation rule that can tolerate $f$ Byzantine workers among the $n$ workers (Blanchard et al., 2017). For centralized attack there is 1 attacker and $n-1$ non-Byzantine workers. For DBA there are $f$ distributed attackers and $n-f$ non-Byzantine workers. The total number of poisoned pixel amounts are kept the same.

**Multi-Krum** To meet the assumption that $2f+2 < n$, we set ($n = 10$, $f = 3$) for LOAN and ($n = 12$, $f = 4$) for image datasets. The Multi-Krum parameter $m$ is set to $m = n - f$. For Tiny-imagenet we decrease the poison ratio into 5/64 for both attacks. Other parameters are the same as described in the paper.

For CIFAR and Tiny-imagenet, we find that DBA is more effective as shown in Fig. 20. For LOAN and MNIST, both attacks don't behave well. We believe the reason can be explained by the fact that LOAN and MNIST are simpler tasks and benign clients quickly agree on the correct gradient direction, so malicious updates are more difficult to succeed.

**Bulyan** We use Bulyan based on the Byzantineresilient aggregation rule Krum (Blanchard et al., 2017). To meet the assumption that $4f + 3 <= n$, we set ($n = 15$, $f = 3$) for LOAN and ($n = 20$, $f = 4$) for image datasets.

For CIFAR, DBA is more effective as shown in Fig. 21. For other datasets, both attacks fail. However, we note that our distributed and centralized backdoor attacks are not optimized for Byzantine setting. We believe its worthwhile to explore the distributed version of other new attack algorithms, e.g. (Baruch et al., 2019) that manipulates its update to mitigate Krum and Bulyan defenses.

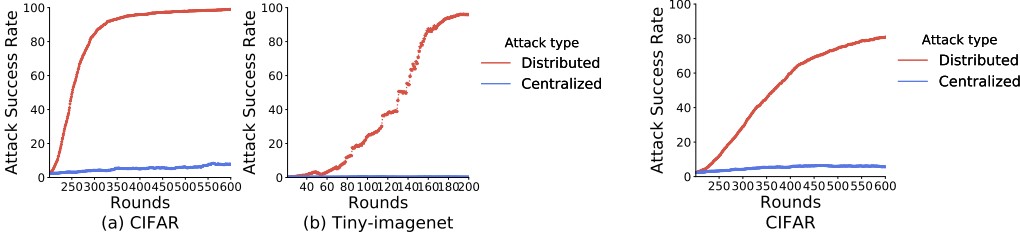

Figure 20: Multi-Krum                    Figure 21: Bulyan

In summary, Multi-Krum and Bulyan have stricter assumptions on the proportion of attackers than RFA and FoolsGold. In addition, while RFA and FoolsGold still assign potential outliers with extreme low weights, Krum (Multi-Krum, Krum-based Bulyan) directly removes them, making it impossible to inject backdoors if the malicious updates are obviously far from the benign updates. The centralized attack for four datasets totally fails under Multi-Krum and Bulyan while DBA can still succeed in some cases.

### A.7    MORE DETAILS ON SOFT DECISION TREE

Frosst & Hinton (2017) proposed Soft Decision Tree which distills a trained neural network by training with data and their soft targets that are the predictions of the neural network over classes. Trained with gradient descent, every inner node has a learned filter and a bias to make binary decision and the leaf node has a learned distribution. To some extent we can use the filter value to reflect the importance of every feature in internal nodes. We learn soft decision trees from the clean neural network and DBA poisoned neural network of LOAN and MNIST and they all achieve about 90% test accuracy on main and backdoor tasks.

If we look at the third node in the forth layer in Fig.22.(b), the potential classifications are only 2 and 0, thus its filter is simply learning to distinguish these two digit. With extreme dark color in the area of the global pattern, which means these pixels correspond to small value in filter, this inner node will make leftmost branch decision into target label 2 when triggered by the global pattern because the probability is lower than 0.5. Taking an opposite example, the leftmost node in second layer has extreme white color in the area of the global pattern, which means these pixels correspond to large value of filter and will contribute to make rightmost branch decision if encountering the global pattern. Moreover, clean images won't trigger the filters in backdoor pattern area and the major digit shape in center dominates the decision route, like examples in Fig.24.(b). Comparing Fig.22.(a)(b), the trigger area after poisoning becomes much more significant for decision making.

Soft Decision Tree provides insights into the neural network and give explainable classification decisions. Examples of the decision routes in inference time for clean and poisoned input data are given for MNIST in Fig.25 and in Fig.24. We find that the poisoned model already starts to misbehave from the top node of the tree.

We also run 10000 poisoned and clean samples for the LOAN clean and poison models to study the sample-wise importance based on the filter value multiplied by the input feature value in Fig.23.(b)(c). With this local importance metric, the original low importance feature indeed becomes salient in poisoned model with poisoned input.

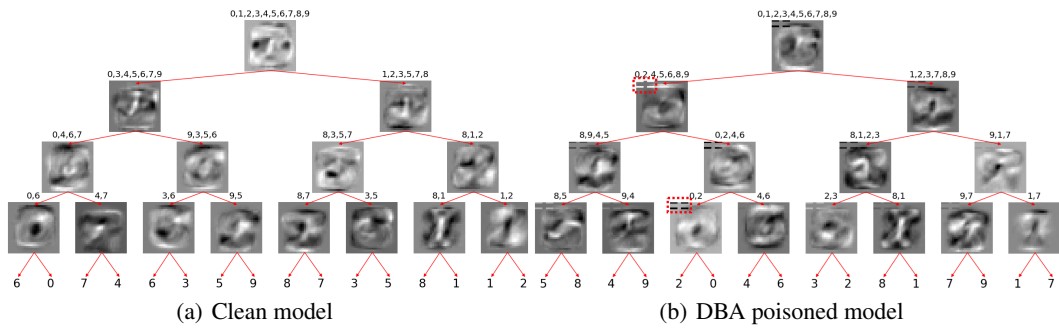

(a) Clean model           (b) DBA poisoned model

Figure 22: Soft decision tree for MNIST

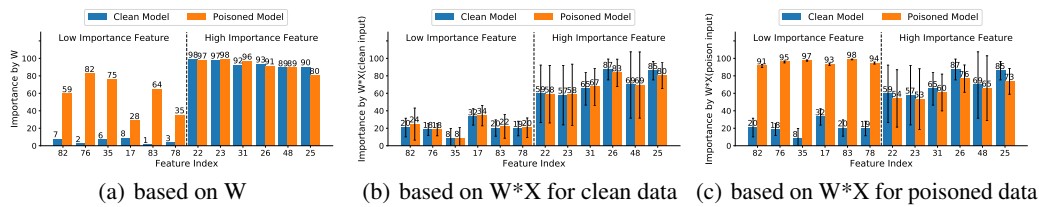

(a) based on W      (b) based on W*X for clean data      (c) based on W*X for poisoned data

Figure 23: Feature importance in soft decision tree for LOAN

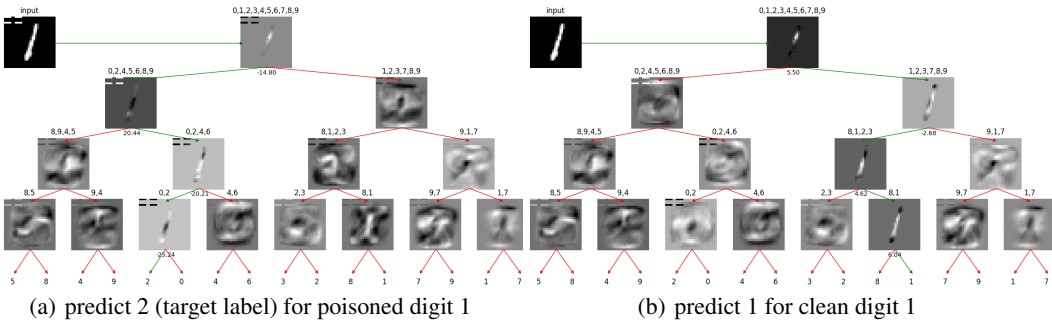

(a) predict 2 (target label) for poisoned digit 1        (b) predict 1 for clean digit 1

Figure 24: Examples for the poisoned MNIST soft decision tree

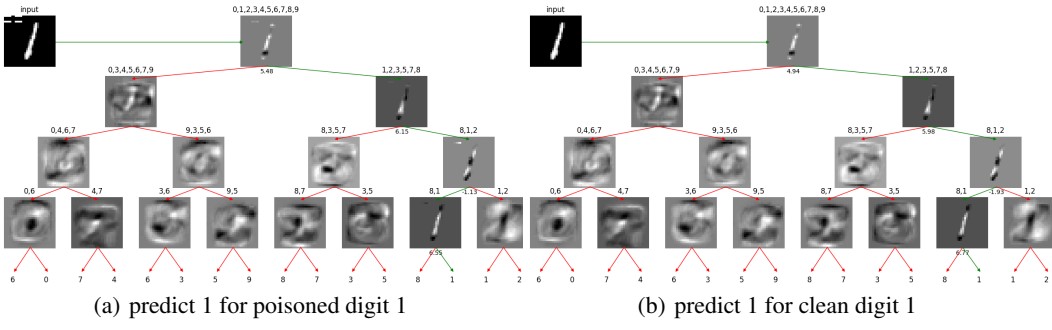

(a) predict 1 for poisoned digit 1          (b) predict 1 for clean digit 1

Figure 25: Examples for the clean MNIST soft decision tree

## A.8 MORE GRAD-CAM RESULTS ON MNIST

We test the global model of MNIST poisoned by DBA under Attack A-M in round 16 of Fig.4 with local backdoored images and global backdoored images. More Grad-cam results are provided in Fig.26 and Fig.27.

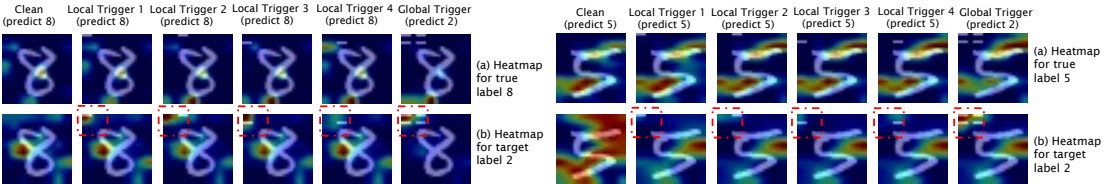

Figure 26: Example of digit 8                  Figure 27: Example of digit 5

## A.9 IMPLEMENTATION DETAILS FOR LOCATION EFFECT EXPERIMENTS

During this process we increases $Shift_y$ and first keeps $Shift_x = Shift_y$. After the rightmost pixel reaches the right edge of images, we fix $Shift_x$ as its largest value, which is X value of the dotted line in Fig.9, and keep increasing $Shift_y$ until the lowest pixel reaches the button edge of the images. $TL$ is the max value among $Shift_x$ and $Shift_y$.

## A.10 DATA DISTRIBUTION EFFECTS FOR TRIGGERS

• By increasing the hypterparameter $\alpha$ in the Dirichlet distribution, we can simulate from non-i.i.d to i.i.d distributions for the image datasets. When evaluated under Attack A-M, Fig.28 shows that DBA-ASR is stable under various distributions, which exhibits the practicability and robustness of DBA when attacking standard FL.

• Data distribution has more influence on the DBA performance under robust aggregation algorithms when calculating distance or similarity between the benign updates and malicious updates. When the training data are non-i.i.d., the updates across the benign participants already appear high diversity so the poisoned update is better concealed among them and less likely to be detected. In our experiments, it's easier for DBA to succeed against RFA and FoolsGold under a more non-i.i.d. data distribution in CIFAR and Tiny-imagenet.

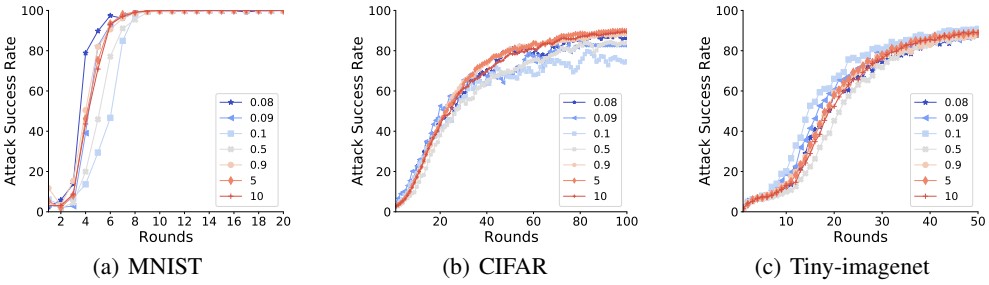

(a) MNIST                  (b) CIFAR                  (c) Tiny-imagenet

Figure 28: Effects of $\alpha$ in Dirichlet data distribution on Attack Success Rate

## A.11 MORE DETAILS ABOUT LOAN DATASETS

The lable distribution is uneven in LOAN, which is shown in Tb.3. The five most important features among the 91 features in LOAN under various classification methods are shown in Tb.4 and the result is consistent.

In Fig. 7, the names for six low importance feature are num_tl_120dpd_2m, num_tl_90g_dpd_24m, pub_rec_bankruptcies, pub_rec, acc_now_delinq, tax_liens; the names six high importance feature are out_prncp,total_pymnt_inv, out_prncp_inv, total_rec_prncp,last_pymnt_amnt, all_util.

Table 3: Financial Dataset Label Distribution

| Loan Status | Number of Examples |
|---|---|
| Fully Paid | 1041952 |
| Current | 919695 |
| Charged Off | 261655 |
| Late (31-120 days) | 21897 |
| In Grace Period | 8952 |
| Late (16-30 days) | 3737 |
| Does not meet the credit policy. Status:Fully Paid | 1988 |
| **Does not meet the credit policy. Status:Charged Off** | 761 |
| Default | 31 |

Table 4: The Five Most Important Features are Similar in Different Classification Methods

| Method | Rank 1st | Rank 2nd | Rank 3rd | Rank 4th | Rank 5th |
|---|---|---|---|---|---|
| Random Forest Classifier | out_prncp | out_prncp_inv | last_pymnt_amnt | recoveries | total_rec_prncp |
| Extra Tree Classifier | out_prncp_inv | out_prncp | total_rec_prncp | last_pymnt_amnt | total_pymnt_inv |
| XGBoost | out_prncp | recoveries | funded_amnt | total_rec_prncp | last_pymnt_amnt |
| Our decision tree | out_prncp | out_prncp_inv | term | recoveries | collection_recovery_fee |

