# OpenReview forum: "DBA: Distributed Backdoor Attacks against Federated Learning"
_ICLR.cc/2020/Conference — Accept (Poster)_

### Official Review · AnonReviewer2 · 2019-10-23
**Official Blind Review #2**

**Rating:** 6

**Review:**

This paper studies backdoor attacks under federated learning setting. To inject a certain backdoor pattern, existing work generate poisoning samples by blending the same pattern with different input samples. Even for federated learning where the adversary can control multiple parties, such as [1], all parties still use the same global backdoor pattern to generate poisoning samples locally. On the contrary, in this work, they decompose the global pattern into several small local patterns, and each adversarial party only uses a local pattern to generate poisoning samples. In their evaluation, they show that the backdoor attacks generated in this way are more effective, resilient to benign model parameter updates, and also survive better against existing defense algorithms against attacks in federated learning settings.

I think the topic studied in this paper is very important and meaningful, and I am convinced that by decomposing a global pattern into several smaller local pieces, the model parameter updates computed by each party should be more similar to benign updates and thus can better bypass the defense algorithms. Meanwhile, the evaluation is pretty comprehensive and it is good to see that the conducted backdoor attacks are effective. However, when there is no defense deployed in the training process, it is not intuitive to see why the proposed attack is more effective and persistent than the centralized attack, given that a smaller trigger usually results in a worse attack performance. Thus, I would like to see more possible explanation on it. Specifically, I have the following questions for clarification:

1. For the evaluation of DBA, I assume that there are 4 adversarial parties, controlling each of the 4 local triggers. When using centralized attacks, are there still 4 adversarial parties, although they share the same global trigger, or if there is only 1 adversarial party?

2. To evaluate A-S setting, I understand that it may be tricky to enable a fair comparison between the centralized attack and DBA. However, one explanation of why DBA is more persistent in this case is because the adversarial parameter updates happen 4x times compared to the centralized attack. Therefore, another baseline to check is to conduct centralized attacks with the same number of times as DBA, but each update includes 1/4 number of poisoning samples, so that the total number of poisoning samples included to compute the gradient update still stays the same.

3. Can the authors show if the decomposition is also useful for trigger patterns that are not necessarily regular shapes? For backdoor attacks, a line of work studies physical triggers, e.g., glasses in [2]. It is not natural to decompose such kind of patterns into several smaller pieces, unless the performance is significantly boosted.

4. Can the authors show concrete examples on how the attacks are generated? The details are especially unclear on LOAN. Specifically, which features are perturbed, what are the values assigned as the trigger, and what is the corresponding target label?

[1]  Bagdasaryan et al., How to backdoor federated learning.
[2] Chen et al., Targeted Backdoor Attacks on Deep Learning Systems Using Data Poisoning.

------------
Post-rebuttal comments

I appreciate the authors' great effort to address my concerns! I think the evaluation in the current version of the paper is pretty comprehensive and provides a valuable study, and I am happy to raise my score accordingly.
-------------

**Experience Assessment:**

I have published in this field for several years.

**Review Assessment: Checking Correctness Of Derivations And Theory:**

I carefully checked the derivations and theory.

**Review Assessment: Checking Correctness Of Experiments:**

I carefully checked the experiments.

**Review Assessment: Thoroughness In Paper Reading:**

I read the paper thoroughly.

---

> ### Author Response · Authors · 2019-11-15
> **Reply to Reviewer 2**
>
> We thank the reviewer for the valuable review comments and suggestions! Please find our point-by-point response as follows.
>
> Q1: Is there only 1 adversarial party?
> A1: There is only one adversarial party in centralized attack. But we make sure that the total injected triggers (e.g., modified pixels) of DBA attackers is close to and even less than that of the centralized attacker.  We stressed this setup in Section 3.2. That is, the ratio of the global trigger of DBA pixels to the centralized is 0.992 for LOAN, 0.964 for MNIST, 0.990 for CIFAR and 0.991 for Tiny-imagenet.
>
> Q2: What’s the result for centralized attacks with the same number of scaling times as DBA, but each update includes 1/4 number of poisoning samples?
> A2: Following your suggestion, we conducted two sets of new experiments.
> 1. Change the poison ratio into 1/4: We decrease the fraction of backdoored samples added per training batch into 1/4.
> 2. Change the data size into 1/4: We divide the local dataset into 4 parts and use 1/4 dataset for each update and keep the poison ratio unchanged.
> We have included the results and discussion in Appendix A.4 of the revised version.
>
> Q3: If the decomposition is also useful for trigger patterns that are not necessarily regular shapes?
> A3: It’s also useful for irregular shape triggers.
> 1. We study the irregular pixel logo ‘ICLR’ for three image datasets. Specifically, we use ‘ICLR’ as the global trigger pattern and decompose it into ‘I’, ‘C’, ‘L’, ‘R’ for local triggers.
> 2. We also use the physical trigger glasses (Chen et al.,2017) on Tiny-imagenet and decomposed the pattern into four parts.
> The results are in Appendix A.3 of our revised version. DBA is also more effective and this conclusion is consistent in different colors of glasses.
>
> Q4: Can the authors show concrete examples on how the attacks are generated? The details are especially unclear on LOAN.
> A4: We note that we have mentioned our attack formulation and algorithm in Section 2.2;
> We have also provided more details about LOAN dataset and how we attack in Appendix A.1.

---

### Official Review · AnonReviewer1 · 2019-10-24
**Official Blind Review #1**

**Rating:** 8

**Review:**

The authors introduce the idea of distributed backdoor attacks in the FL framework, in which the dishonest participants in FL add local triggers to their training data to influence the global model to classify triggered images in a desired way. They show empirically that the learned models then are more likely to be successfully forced to misclassified images in which all the local triggers are present at test time, than are models learned using centralized backdoor attacks, where all attackers use the same trigger pattern (one of the same size as the concatenation of the local triggers, to be fair in the comparison). They then demonstrate that because the local triggers cause smaller corruptions in the model coefficients, these distributed attacks survive robust FL training algorithms (namely FoolsGold, and a recent robust regression based method) more often than centralized attacks. Similar experiments are conducted on the Loan text dataset, using appropriate analogs of local triggers, with similar results.

The paper contributes a novel model for conducting backdoor attacks in the FL setup, and shows that this model is more successful at attacking when training using robust FL algorithms than the standard centralized backdoor attack model. I lean towards accept, as this is a realistic attack model, and as such can further stimulate research into the robustification of FL model aggregation algorithms.

**Experience Assessment:**

I do not know much about this area.

**Review Assessment: Checking Correctness Of Derivations And Theory:**

N/A

**Review Assessment: Checking Correctness Of Experiments:**

I assessed the sensibility of the experiments.

**Review Assessment: Thoroughness In Paper Reading:**

I read the paper thoroughly.

---

> ### Author Response · Authors · 2019-11-15
> **Reply to Reviewer 1**
>
> Thank you for your appreciation of our work!

---

### Official Review · AnonReviewer3 · 2019-10-25
**Official Blind Review #3**

**Rating:** 6

**Review:**

This paper proposes a distributed backdoor attack strategy, framed differently from the previous two main approches (1) the centralised backdoor approach and (2) the (less discussed in the paper) distributed fault tolerance approach (often named "Byzantine").

The authors show through experiments how their attack is more persistent than centralised backdoor attack.
The authors also compare two aggregation rules for federated learning schemes, (Fung et al 2018 & Pillutla et al 2019), suggesting that both rules are bypassed by the proposed distributed backdoor attack.

Strength:

what I found most interesting in the paper is Section 3.4, presenting an appreciable attempt to "interpret" poisoning. Together with Section 4.
This kind of fine-grained analysis of poisoning is highly needed.

Weakness:

in section 3.3, the authors compare against RFA and take what is claimed in Pillulata et al as granted (that RFA detects more nuanced outliers than the wort-case of the Byzantine setting (Blanchard et al 2017) ). In fact, there is more to the Byzantine setting than that, see e.g. Draco (Chen et al 2018 SysML), Bulyan (El Mhamdi et al 2018 ICML) and SignSGD (Bernstein et al 2019 ICLR) which have proposed more sophisticated approches to distributed robustness.
Since this paper is about distributed robustness and distributed attacks, it would be very informative to the community to illustrate DBA attack on these methods to have a more compelling message.

post rebuttal: thank your for your detailed reply, I acknowledge your new comparisons with the distributed robustness mechanisms of Krum and Bulyan, too bad time was short to compare with the other measures such as Draco and SignSGD.

**Experience Assessment:**

I have published in this field for several years.

**Review Assessment: Checking Correctness Of Derivations And Theory:**

I carefully checked the derivations and theory.

**Review Assessment: Checking Correctness Of Experiments:**

I assessed the sensibility of the experiments.

**Review Assessment: Thoroughness In Paper Reading:**

I read the paper thoroughly.

---

> ### Author Response · Authors · 2019-11-15
> **Reply to Reviewer 3**
>
> Thanks so much for your valuable review comments!
>
> Following your suggestion, we evaluated the Byzantine settings Multi-Krum (Blanchard et al 2017) and Bulyan (El Mhamdi et al 2018 ICML). For both DBA and centralized attack we use the aggregation rule that can tolerate f Byzantine workers among n workers (Blanchard et al 2017). For centralized attack there is 1 attacker and n-1 non-Byzantine workers. For DBA there are f distributed attackers and n-f non-Byzantine workers.  The total number of poisoned pixel amounts are kept the same.
>
> 1. Multi-Krum
> - To meet the assumption that 2f + 2 < n, we set  (n=10, f=3) for loan and (n=12, f=4) for image datasets. The Multi-Krum parameter m is set to m=n-f. For Tiny-imagenet we decrease the poison ratio to 5/64 for both attacks. Other parameters are the same as described in the paper.
> - For CIFAR and Tiny-imagenet, we find that DBA is more effective.
> - For LOAN and MNIST, both attacks don’t behave well. We believe the reason can be explained by the fact that Loan and MNIST are simpler tasks and benign clients quickly agree on the correct gradient direction, so malicious updates are more difficult to succeed.
>
> 2. Bulyan
> - We use Bulyan based on the Byzantine–resilient aggregation rule Krum. To meet the assumption that 4f + 3 <= n, we set  (n=15, f=3) for loan and (n=20, f=4) for image datasets.
> - For CIFAR, DBA is more effective.
> - For other datasets, both attacks fail. However, we note that our distributed and centralized backdoor attacks are not optimized for Byzantine setting. We believe it’s worthwhile to explore the distributed version of other new attack algorithms, e.g. A Little Is Enough (Baruch et al 2019) that manipulates its update to mitigate Krum and Bulyan defenses.
>
> In summary, Multi-Krum and Bulyan have stricter assumptions on the proportion of attackers than RFA and FoolsGold. In addition, while RFA and FoolsGold still assign potential outliers with extreme low weights, Krum (Multi-Krum, Krum-based Bulyan) directly removes them, making it impossible to inject backdoors if the malicious updates are obviously far from the benign updates. The centralized attack for four datasets totally fails under Multi-Krum and Bulyan while DBA can still succeed in some cases. We have included these results in Appendix A.6 of the revised version.

---

### Decision · Program_Chairs · 2019-12-19

**Decision:**

Accept (Poster)

**Comment:**

Thanks for the discussion, all. This paper proposes an attack strategy against federated learning. Reviewers put this in the top tier, and the authors responded appropriately to their criticisms.